# Progastrin: An Overview of Its Crucial Role in the Tumorigenesis of Gastrointestinal Cancers

**DOI:** 10.3390/biomedicines12040885

**Published:** 2024-04-17

**Authors:** Rodanthi Fioretzaki, Panagiotis Sarantis, Nikolaos Charalampakis, Konstantinos Christofidis, Adam Mylonakis, Evangelos Koustas, Michalis V. Karamouzis, Stratigoula Sakellariou, Dimitrios Schizas

**Affiliations:** 1First Department of Surgery, National and Kapodistrian University of Athens, Laikon General Hospital, 11527 Athens, Greece; rodanthifioretzaki@gmail.com (R.F.); adam.mylonakis@gmail.com (A.M.); schizasad@gmail.com (D.S.); 2Department of Biological Chemistry, School of Medicine, National and Kapodistrian University of Athens, 11527 Athens, Greece; panayotissarantis@gmail.com (P.S.); mkaramouz@med.uoa.gr (M.V.K.); 3Department of Medical Oncology, Metaxa Cancer Hospital of Piraeus, 18537 Piraeus, Greece; nick301178@yahoo.com; 4First Department of Pathology, School of Medicine, National and Kapodistrian University of Athens, 11527 Athens, Greece; konstantinos.christofidis@gmail.com (K.C.); sakellarioustrat@yahoo.gr (S.S.)

**Keywords:** progastrin, hPG80, gastrointestinal cancers, gastrin, colorectal cancer

## Abstract

Defining predictive biomarkers for targeted therapies and optimizing anti-tumor immune response is a main challenge in ongoing investigations. Progastrin has been studied as a potential biomarker for detecting and diagnosing various malignancies, and its secretion has been associated with cell proliferation in the gastrointestinal tract that may promote tumorigenesis. Progastrin is a precursor molecule of gastrin, synthesized as pre-progastrin, converted to progastrin after cleavage, and transformed into amidated gastrin via biosynthetic intermediates. In cancer, progastrin does not maturate in gastrin and becomes a circulating and detectable protein (hPG80). The development of cancer is thought to be dependent on the progressive dysregulation of normal signaling pathways involved in cell proliferation, thus conferring a growth advantage to the cells. Understanding the interaction between progastrin and the immune system is essential for developing future cancer strategies. To that end, the present review will approach the interlink between gastrointestinal cancers and progastrin by exploring the underlying molecular steps involved in the initiation, evolution, and progression of gastrointestinal cancers. Finally, this review will focus on the clinical applications of progastrin and investigate its possible use as a diagnostic and prognostic tumor circulating biomarker for disease progression and treatment effectiveness, as well as its potential role as an innovative cancer target.

## 1. Introduction

Progastrin is a precursor molecule of gastrin. Gastrin is a peptide hormone produced mainly in the gastric antrum by endocrine G-cells, with lower expression in the duodenum and pituitary gland. Gastrin plays a crucial role in the maintenance and functioning of the gastric mucosa in the stomach. It was initially synthesized as a precursor peptide known as pre-progastrin, consisting of 101 amino acids. After cleavage of a signal peptide, it gets converted to progastrin. The 80 amino acid precursor, progastrin, is then transformed in the endoplasmic reticulum to amidated gastrin via biosynthetic intermediates [1,2].

Under normal conditions, progastrin is not detected in the blood of healthy people. In tumor cells, progastrin is not maturated into gastrin but is released from the cells. Progastrin becomes a circulating protein, hPG80, which can be detected in cancer patients [3,4,5,6].

In 1990, Bardram made the first link between progastrin and tumors. He evaluated the presence of circulating progastrin in the blood of patients with Zollinger–Ellison syndrome. He demonstrated that a small amount of progastrin could predict a malignant condition at an early stage of the disease. He noted that the total progastrin product better reflects tumor gastrin synthesis than conventional measures of alpha-amidated gastrin [7]. The evidence showing that progastrin could be identified and quantified in the blood of colorectal cancer (CRC) patients was demonstrated by Siddheshwar et al. as early as 2001 [8]. The hypothesis of progastrin’s functional autocrine/paracrine activity in tumor cells was supported after Singh et al. showed that progastrin was secreted from these cells when cultivated in vitro [9]. 

The detection and quantification of progastrin in the blood of colorectal cancer patients, demonstrated by Siddheshwar et al. in 2001 [8], was subsequently confirmed by Prieur et al. in 2017 [10]. Progastrin levels in the blood of CRC patients were measured before and after surgery by Konturek et al. in 2002 [11]. They discovered that rates were higher before surgery than controls and returned to normal after surgery.

It was also demonstrated that progastrin had to be released from tumor cells in order to exert its oncogenic functions. hPG80 was found in the blood of 11 different types of cancer patients, demonstrating the role that progastrin may have in cancer pathogenesis [6]. This implies the significant influence that progastrin may exert on tumors, laying the groundwork for progastrin to be considered a new target in the fight against cancer.

## 2. hPG80 Gene and Receptor

In humans, progastrin is encoded by the *GAST* gene located on chromosome 17 (17q21) [12]. Previous research with anti-gastrin antibodies suggested that gastrin could act as colon cancer’s autocrine growth factor. The growth of human colon cancer cells that express high levels of gastrin mRNA may be influenced by the expression of the gastrin (GAST) gene in a physiologically relevant manner. As the gastrin gene is expressed in >60% to 80% of human colon cancers, it is reasonable to assume that a sizeable portion of these tumors may strictly depend on the *GAST* gene products [13]. Listed below are the main signaling pathways where they activate the *GAST* gene (Figure 1).

### 2.1. Wnt/β-Catenin Signaling Pathway

In the majority of cases, the first incident that leads to colorectal cancer is the constitutive stimulation of the Wnt/β-catenin pathway resulting from mutations in either *the β-catenin* or *APC* (Adenomatous polyposis coli) coding gene. Induction mutations in normal intestinal stem cells are sufficient to initiate tumorigenesis. The Wnt oncogenic pathway activates the *GAST* gene, a β-catenin/Tcf-4 downstream target [14].

### 2.2. K-Ras, MEK-ERK Signaling Pathways

Progastrin and the K-Ras oncogenic pathway mutations have also been interlinked. GAST mRNA levels were significantly higher in CRC tissues with K-Ras mutations than in wild-type (WT) K-Ras. GAST expression is affected by K-Ras via the Raf-MEK-ERK signal pathway, leading to the stimulation of the GAST gene. Because the K-Ras and the Wnt pathways both upregulate *GAST* gene expression, these two pathways may cooperate in regulating hPG80 expression. Chakadar et al. stated that combining oncogenic β-catenin and K-Ras overexpression resulted in a notable activation of the *GAST* gene promoter. hPG80 expression is regulated by these two pathways in around 50% of human CRC tumors [15].

### 2.3. PI3K/Akt, NF-κB, SMAD4

The PI3K/Akt pathway has been mainly implicated in intestinal epithelial proliferation in various tumor cells, like those of colonic origin. Elevated levels of Akt phosphorylation have been identified and linked with raised cell proliferation [16,17]. Inhibition of the PI3K/Akt pathway by a specific PI3K inhibitor caused apoptosis in various human colonic cancer cells in vitro and in vivo [17,18]. Indeed, in the colonic mucosa of MTI/G-Gly [mice that overexpress progastrin truncated at glycine-72(G-Gly)] and hGAS mice (transgenic mice with increased plasma progastrin levels), the p85 regulatory subunit of PI3K and Akt was upregulated. These results supported the PI3K/Akt pathway’s vital involvement in regulating colonic proliferation induced by non-amidated gastrins, such as progastrin and G-Gly. *GAST* gene activation was also abolished via PI3K inhibition.

Another principal signaling messenger controlled by hPG80 is NF-kB. Its role in the processes behind hPG80’s anti-apoptotic action has been established in pancreatic cancer cells overexpressing the *GAST* gene [19].

*GAST* gene promoter stimulation was further enhanced or suppressed via the co-expression of WT *SMAD4* or a dominant negative mutant of *SMAD4*. *SMAD4* is essential for transcriptional and antiproliferative responses to TGF-β [19].

### 2.4. Progastrin Receptor

The identification of progastrin-binding sites has been a subject of recent research. Recombinant iodinated human progastrin defined the first high-affinity binding sites in intestinal epithelial cells (IECs). The affinity ranged from 0.5 to 1 nM, which was within the range of a receptor [20]. Competition with unlabeled cold PG—but not with glycine extended gastrin (Gly-G-17) or amid-17 gastrin—verified binding specificity. The conventional cholecystokinin-2 receptor (CCK-2R) did not affect binding. These studies revealed the existence of a binding site for progastrin, distinct from amidated gastrin and prolonged gastrin to glycine 17. The sequence of progastrin interacting with this receptor is probably positioned at amino acid residue 26, but its identity remains unclear [21].

Annexin II, a cell membrane protein, has been identified as a possible receptor of PG in gastric cancer cells. Annexin II is a partial mediator of the effect of PG and is able to mediate NF-κΒ and β-catenin upregulation in response to PG in mice and HEK-293 cells. Annexin II could further play a role in the clathrin-mediated endocytosis of progastrin [21]. Because of its widespread presentation in several forms of cancer, it has also been investigated as a prognostic marker in gastric cancer. The expression of Annexin II has been associated with poor tumor differentiation, larger tumor size, and advanced prognostic stage [21,22].

Another possible candidate is the G protein-coupled receptor 56 (GPCR56), expressed on both colon and cancer stem cells [23]. It was proposed that GPR56 may play a role in the progastrin-induced promotion of CRC. It was discovered that GPR56 is expressed in uncommon colonic crypt cells that lineage certain colonic glands, which is consistent with GPR56 identifying long-lived colonic stem cells. In transgenic mice that overexpressed human progastrin, GPR56 was upregulated. Although colonic organoids cultured from GPR56 mice were resistant to PG progastrin, wild-type colonic organoids grew and survived with recombinant human progastrin. In vitro, it has been shown that progastrin is directly bound to and stimulates the proliferation of colon cancer cells that express the GPR56 receptor as well as the CCK2R. In the presence of progastrin, colonic mucosal proliferation was inhibited in vivo, and apoptosis was elevated. In the (AOM) CRC mouse model, loss of GPR56 also inhibited progastrin-dependent colonic crypt fission and colorectal carcinogenesis [23].

Progastrin signal transduction by the unidentified progastrin receptor involves several intracellular intermediates linked to tumorigenesis. The progastrin receptor activates various signaling pathways, demonstrating unique characteristics in a way that is difficult to identify.

## 3. Progastrin Role on Colorectal Cancer Cells

Colorectal cancer (CRC) can arise from the progressive accumulation of multiple genetic and epigenetic aberrations within cells. There are three major pathways associated with CRC pathogenesis: chromosomal instability (CIN), CpG Island Methylator Phenotype (CIMP), and Microsatellite Instability (MSI). According to Fearon [24], the first step of CRC tumorigenesis is the constitutive activation of the Wnt/ß-catenin oncogenic pathway induced by the mutation of the Adenomatous Polyposis Coli gene (*APC*), the most frequent, or *ß-catenin*, followed by the mutational activation of the oncogene *KRAS* and the inactivation of the tumor suppressor gene *TP53*. Then, cancer progresses through the stepwise accumulation of multiple genetic and epigenetic aberrations, leading to invasive and metastatic tumors.

In addition, neo-angiogenesis supplies metabolic substrates to cancer cells orchestrating a microenvironment that suppresses productive anti-tumor immunity [25,26].

### 3.1. Progastrin as a Factor in Tumorigenesis

Singh et al. 1996 gave the first evidence supporting the crucial role of gastrin gene expression in human colon cancers [13]. They investigated the impact of gastrin gene expression on the progress of colon cancer cells by examining the effect of gastrin antisense (AS) RNA expression, which inhibits progastrin production. To carry out the study, they transfected three human colon cancer cell lines expressing different levels of gastrin mRNA with either AS or control vectors and conducted growth studies in vitro and in vivo. The research indicates that AS clones derived from cell lines expressing gastrin have reduced potential for proliferation and tumorigenesis compared to control clones. Moreover, the study suggests that high levels of gastrin in hypergastrinemic mice are associated with an elevated risk of colorectal carcinoma. Finally, the analysis of the antiproliferative effects observed in AS gastrin RNA expression indicates that progastrin may play a significant role as a growth factor since its concentration reduction produced specific results [13].

Holland et al. demonstrated that gastrin extended with glycine acts as a trophic factor in non-transformed cells [27]. Another study showed that glycine-extended gastrin stimulates the growth of Human Embryonic cells (HEKs) and human colon cancer cells in vitro [28]. These results support the vital role of progastrin maturation products in the tumor growth of colon cancer. A study conducted by Wang et al. created the first permanent transgenic mouse model of human progastrin and amidated gastrin overexpression, implying the existence of another cholecystokinin (CCK)/gastrin receptor in the colon. The increased colonic proliferation in hypergastrinemic mice supports the hypothesis that high gastrin levels are linked to an increased risk of colorectal cancer [29].

Azoxymethane (AOM) treatment of mice overexpressing progastrin caused a considerable rise in tumor growth [30]. Studies with transgenic mice that overexpress progastrin provide strong evidence that high levels of progastrin can increase the risk of developing carcinoma by significantly augmenting the carcinogenic potential of chemical carcinogens [30].

In another experiment by Koh et al., mice with an *APC* gene mutation were crossed with gastrin-deficient mice [31]. An allele of the gene in the APCmin/+ mouse acquires a mutation resulting in its inactivation, causing intestinal tumorigenesis to begin, manifesting as spontaneous adenomas and eventually adenocarcinomas. Moreover, in the deficient mice, a marked decrease was observed in the number of polyps and a reduced rate of polyp proliferation.

Pannequin et al. [32] and Prieur et al. [33] used another mouse model with a different mutation in the *APC* gene, APC∆/+. These mice spontaneously developed adenomas and adenocarcinomas, with more tumors in the colon. Both studies altered progastrin by treating mice with siRNA or a neutralizing anti-progastrin antibody. Notably, in all experiments, the inhibition or neutralization of progastrin reduces the number of tumors.

### 3.2. Progastrin Is Essential for the Survival of Tumor Stem Cells

It was crucial to determine whether progastrin’s in vivo impact on the development of intestinal tumors might involve the control of cancer stem cells (CSCs). The finding that progastrin is expressed in CD133^pos^ colorectal cancer cells, which exhibit some of the phenotypic traits of cancer stem cells, suggests that it may play such a role [34]. As demonstrated, β-catenin is activated in response to progastrin. The impact of progastrin on colon cancer cells and colonic crypts of mice was studied, considering that putative stem cell markers such as CD44, DCLK1, Lgr5, and CD133 are the target genes of β-catenin/Tcf/Lef transcriptional factors. Endocrine progastrin was found to enhance the expression of these stem cell markers, both in vitro and in vivo [21,35]. The down-regulation of gastrin gene expression in DLD1 cells (CRC cell line isolated from the large intestine of a colon adenocarcinoma patient) significantly reduced the expression of CD133, resulting in loss of tumorigenic potential of the cells in vivo. Thus, the down-regulation of progastrin may attenuate the tumorigenic potential of cancer stem cells [36].

Giraud et al. investigated progastrin’s crucial function in cancer stem cells [37]. They demonstrated that colorectal cancer cells cultured in an enriched environment for cancer stem cells greatly increased progastrin expression at mRNA and protein levels.

It was later discovered that progastrin is essential for developing spheres that require a cancer stem cell to begin growing. This suggested that progastrin could control the frequency of cancer stem cells, which was later shown to be true both in vitro and in vivo. Subsequent studies by Prieur et al. showed that migration and invasion—two traits of cancer stem cells—significantly impact both in vitro and in vivo. Progastrin also functions as a survival factor for CSCs. Neutralizing antibodies can reduce the frequency of CSCs both in culture and in mice grafted with human colorectal cancer cells [6,33,37].

### 3.3. Progastrin Decreases Apoptosis

As already mentioned, progastrin is able to stimulate the proliferation of colon cancer cells. Wu et al. have shown that progastrin may directly or indirectly mediate the observed growth effects in these cells. Treatment with progastrin of intestinal epithelial cells led to a significant loss of caspase-3 and caspase-9 activation and reduced DNA fragmentation. Thus, the effect of progastrin on cell survival derives from raised proliferation and reduced apoptosis. These results suggest the possibility that progastrin exerts direct anti-apoptotic effects on the target cells. Additionally, the upregulation of cytochrome c oxidase Vb levels and mitochondrial synthesis of ATP may also contribute to the increase in the growth of cancer cells in response to PG [38].

### 3.4. Progastrin Is a Pro-Angiogenic Factor

Angiogenesis is essential for tumor progression and metastasis. A study by Najib et al. correlated progastrin overexpression with increased vascularization. In vitro treatment with progastrin increased endothelial cell proliferation and their ability to form capillary-like structures. Moreover, it was demonstrated that progastrin increased endothelial cell migration and enhanced their permeability by phosphorylation of vascular endothelial (VE)-cadherin, p125-FAK, paxillin, and induction of actin remodeling. In vivo, preventing the production of progastrin with shRNA (cells stably transduced with a shRNA against PG) in xenograft cells in nude mice resulted in a decrease in tumor growth, endothelial permeability, and neovascularization [39].

Even if data on the interplay among hypoxia and gastrin are limited, it has been proved that hypoxia upregulates the promoter activity of the gastrin gene in GI cancer cell lines, and gastrin counterbalances hypoxia by stimulating angiogenesis in vitro and in vivo [40,41]. The work of Laval et al. [42] provided the first evidence of an association involving progastrin overexpression and hypoxia, demonstrating that in vivo, increased circulating concentrations of progastrin provide a physiological advantage to hGAS mice against systemic hypoxia. Later, in 2017, Prieur et al. demonstrated that progastrin expression, under hypoxic conditions, is stimulated in vitro [33]. These results prove that progastrin, recognized as a growth factor, is also a potent pro-angiogenic factor.

### 3.5. Progastrin Regulates Adhesions and Tight Junctions

The dysregulation of cellular adhesion plays a critical role in malignant transformation and the metastasis of cancer cells [43,44]. It was confirmed that inhibiting hPG80 secretion with an antisense (AS) construct directed against progastrin mRNA restores membrane localization of tight and adherent junction constituent proteins in the CRC cell line DLD-1 [45]. Thus, hPG80 influences cell contact integrity.

### 3.6. Progastrin and Tumor-Reactive Stroma

Progastrin activates cancer-associated fibroblasts (CAFs) and contributes to the interaction between tumor epithelial cells and stromal fibroblasts. The tumor-reactive stroma contains CAFs, which increase the aggressiveness and motility of epithelial cancer cells. Although the importance of tumor-stroma cross-talks in cancer development is well recognized, it is still unclear how “normal” local fibroblasts become pro-invasive CAFs and how CAFs contribute to the epithelial-stroma conversation. Progastrin, recognized as a growth factor for colon cells, is absent in healthy colons but present in colorectal polyps and tumors. Fénié et al. evaluated the migration of colon epithelial cancer cells—expressing or not progastrin—in the presence or absence of fibroblasts in transgenic mice and human colon fibroblast cell lines [46]. While there was no difference between the two epithelial cell lines in the presence of fibroblasts, progastrin-expressing epithelial cells showed higher migratory capacity. Progastrin is known to activate cancer-associated fibroblasts (CAFs) and act as a mediator between tumor epithelial cells and stromal fibroblasts.

Additionally, in vitro studies have shown that progastrin stimulates the proliferation of colonic myofibroblasts, while in vivo experiments involving its transgenic overexpression reveal an increase in the number of myofibroblasts in the colonic mucosa. Colonic pericryptal myofibroblasts and colonic epithelial proliferation are elevated in progastrin-overexpressing hGAS animals. Progastrin promotes colonic myofibroblast growth in vitro via the Protein Kinase C pathway. Hence, progastrin affects both stromal and epithelial compartments to raise colonic susceptibility to carcinogenesis [47].

## 4. Progastrin Role in Other Gi Cancers

### 4.1. Progastrin and Gastric Cancer

Recognizing the stomach’s tissue and stem cell structure is complex because of differences in the anatomic location (antrum, corpus, and fundus). It is true that despite significant advancements in our understanding of the stem cell niche in various gastric areas, it remains challenging to determine which, if any, of these stem cells give rise to cancer stem cells. In a mouse Smad4/PTEN deletion model, has been explored the function of Lgr5+ stem cells in carcinogenesis. It has been discovered that gene deletion in Lgr5+ stem cells leads to the formation of adenoma and invasive intestinal-type gastric cancer (GC) after three months, mainly in the antrum. On the other hand, some cell populations can operate as functional or reserve stem cells and can be summed in the event of tissue injury. The stem cell marker Troy is expressed at the gland base by a small subpopulation of fully differentiated stem cells that show plasticity, especially after an injury. Troy+ cells in the gastric corpus were found by Stange et al. to be differentiated yet slowly cycling cells compared to Lgr5+ cells [48,49]. Therefore, Troy and progastrin may have a relation in the regulation of gastric stem cell dynamics and tumorigenesis.

Hayakawa et al. discovered that antral cells expressing the progastrin/gastrin receptor CCK2R are stem cell-like and have a long lifespan [50]. Progastrin increased organoid formation and Lgr5 expression in CCK2R+/Lgr5- cells, indicating that they can differentiate into Lgr5+ stem cells. These data imply that differentiated cells can reactivate earlier developmental pathways if necessary. Research on the small intestine has revealed similar data, indicating that committed enterocyte progenitors have the capability to replace Lgr5+ cells after severe injury [51].

In a case–control study conducted by Amjadi et al., the aim was to determine the diagnostic significance of progastrin serum biomarkers in patients with gastric cancer. The study included 40 untreated patients with gastric cancer and a control group of 42 individuals with no history of gastrointestinal cancer. The level of progastrin serum was measured using an ELISA kit, which showed significantly higher levels in patients with gastric cancer compared to the control group. However, there was no significant correlation between progastrin serum level and tumor clinicopathologic parameters [52].

### 4.2. Progastrin and Liver

Gastrin and progastrin are synthesized in the stomach and metabolized in the liver, but little is known about their levels in various hepatic disorders. The study of Konturek at al. examined 147 patients with chronic hepatitis B, hepatitis C, liver cirrhosis, and age- and sex-matched healthy controls. Patients with cirrhosis had significantly higher plasma levels of progastrin and gastrin as a result of impaired metabolism and increased release from gastric cells due to H. pylori infection [53].

Caplin et al. conducted a study to analyze the expression of CCK-B/gastrin receptor (CCK-BR), progastrin, G-Gly, and G-NH2 in normal liver and primary liver tumors. The findings revealed that the majority of liver tumors express CCK-BR and can process progastrin to gastrin and G-Gly but not to the amidated form. In contrast, the normal liver displays low levels of receptor expression and no expression of precursor forms of gastrin. This expression may be associated with tumor proliferation [54].

Alpha-fetoprotein (AFP) is the most widely used biomarker for HCC prognosis in the advanced stages of the disease but is not helpful in the early stages. Prieur et al. aimed to evaluate the prognostic value of circulating progastrin—alone or combined with AFP in patients with HCC [10]. This study found a significant correlation between progastrin expression and an advanced stage of the disease in patients with HCC. HPG80 can be combined with AFP as a new prognostic biomarker in HCC, especially in patients with negative AFP and early-stage HCC. The liver operates in a state of relative hypoxia, and resected human CRC liver metastases contain a high proportion of hypoxic cells when compared to other tumor types. According to this study, progastrin expression could potentially predict aggressive tumor behavior in patients with colorectal cancer (CRC). In cases where the expression of the gastrin gene is continuously upregulated in primary tumors, CRC cells that disseminate from these tumors could establish metastatic foci in the relatively hypoxic liver microenvironment. One possible explanation for this phenomenon is the resistance to hypoxia-induced cell death mediated by non-amidated gastrin precursors [10].

Due to a wide range of mechanisms that are critical for tumor growth and survival, progastrin can be considered a major tumor promoter. Its principal function is to promote cancer stem cells’ survival and spread them to form metastases, thus making it a potential marker of liver metastasis in colorectal cancer [55].

## 5. Clinical Implications

Progastrin is detected in the blood of cancer patients and is also found in preneoplastic stages like adenomatous polyps. Since it is produced by tumor cells in both the primary tumor and metastasis, it can be used for patient monitoring. Its concentrations in the blood have been found to rise in patients at risk of developing CRC [56]. Progastrin could also be used as a biomarker of liver metastasis in CRC [55]. Regarding progastrin as a therapeutic option, the fact that cancer stem cells require progastrin to survive is critical because, currently, no drug exists that can target cancer stem cells directly [33]. Targeting progastrin also makes tumor cells more sensitive to radiotherapy, which may help radiation treatment to be more effective [57]. Furthermore, chemotherapy causes a significant increase in progastrin in CRC cells [33] (Figure 2).

### 5.1. Monitoring Disease Activity and Treatment Efficacy

Plasma assays of hPG80 in CRC patients reveal that the presence of hPG80 correlates with tumor formation. hPG80 concentrations are elevated in patients at risk of developing the disease and in hyperplastic polyps that have proceeded to cancer [11,56].

hPG80 could be a biomarker of liver metastasis. A recent study has demonstrated a decrease in hPG80 levels following surgery in a group of patients with peritoneal involvement from GI cancers who underwent peri-operative chemotherapy regimens and cytoreductive surgery. The study also established a correlation between hPG80 levels and standard imaging in a cohort of patients with hepatocellular carcinoma (HCC), including patients with alpha-fetoprotein (AFP) < 20 ng/mL. Patients in remission have decreased levels of hPG80 than the patients with active cancer [55].

### 5.2. Antibody Therapy

Prieur et al. presented a work focusing on the innovation of humanized mAbs with a high affinity for native progastrin and therapeutic properties [33]. Initially, the study documented an increase in both plasma and cellular progastrin levels in response to exposure to chemotherapy or hypoxia-inducing conditions in the colon. This highlights the clinical significance of progastrin as a target in advanced-stage colorectal cancer patients receiving therapy. Interestingly, it was shown that humanized anti-progastrin antibodies were able to decrease cell proliferation and increase cell death. Additionally, these antibodies notably reduce CSC self-renewal in vitro and in vivo and impair Wnt signaling and APC mutation-driven tumorigenesis. Finally, the therapeutic potential of the anti-progastrin antibodies is further reinforced by the demonstration that, when combined with chemotherapy, they can increase chemosensitivity and delay relapse after the treatment of in vivo tumors. The above has lately been proven for esophageal, liver, breast, ovarian, and gastric cancers, suggesting that this antibody is a promising multi-cancer drug [6].

### 5.3. Sensitization to Radiotherapy

Many advanced rectal cancers, accounting for approximately 40% of total CRCs, are resistant to radiotherapy (RT). Targeting factors associated with radioresistance to increase the cytotoxic effects of radiotherapy can be an effective approach to enhance the curative resection rate and minimize the likelihood of metastasis [57].

It has been observed that progastrin mRNA expression increases under RT. In vitro, using a shRNA approach, it was demonstrated that the down-regulation of progastrin mRNA expression radiosensitizes CRC cells. The research found that combining progastrin gene down-regulation with RT effectively inhibits tumor progression in vivo. Then, it was observed that when targeting the progastrin gene, cancer cells become radiosensitized by increasing radio-induced apoptosis and up-regulation of the pro-apoptotic pathway JNK. Furthermore, inhibition of progastrin gene expression enhances radiation-induced DNA damage. Moreover, targeting the progastrin gene also inhibits AKT and ERK survival pathways induced by RT to increase radiation-induced apoptosis. The study’s findings underscore the pivotal role of progastrin in radioresistance and provide evidence that progastrin, which is typically overexpressed in colorectal cancer and acknowledged as a pro-oncogenic factor, has the potential to be a promising target for sensitizing resistant rectal tumors to radiation therapy (RT) [57].

These findings suggest that targeting the progastrin gene could potentially improve the efficacy of RT in the treatment of CRC, particularly in cases where tumors have become resistant to radiation. Further research is needed to explore these results’ clinical implications and develop targeted therapies that can effectively inhibit PG expression.

### 5.4. Clinical Trials

The assessment of the value of progastrin as a tumor screening assay and as a monitoring test in patients with GI carcinomas is an area of intensive ongoing trial investigation.

The efficacy of progastrin as a diagnostic and therapeutic monitoring marker in patients treated for peritoneal carcinomatosis from gastrointestinal (GI) malignancies was evaluated in the prospective BIG-RENAPE trial (NCT02823860). Based on the provided information, it appears that the ELISA DECODE LAB test^®^ was used to measure progastrin in 190 GI cancer patients vs. 80 healthy donors. According to the study, the area under the ROC curve of progastrin for cancer diagnosis was 0.87, with a 95% confidence interval of [0.83–0.92]. Progastrin was found to be significantly elevated at inclusion in all GI tumor subtypes (*p* < 0.0001; median 3.08, 95% CI [1.15–7.23]), including colorectal and small bowel (n = 151; median 2.78) and oeso-gastric (n = 33; median 4.75) carcinomas. During monitoring, progastrin levels decreased after neoadjuvant treatment (median value 2.20, n = 23), and this decrease became significant after surgery (*p* < 0.0001, median value 1.57, n = 84). Some patients’ progastrin levels went back to normal, while others did not. The study also observed a trend for better progression-free survival (PFS) in patients who experienced a decline in progastrin levels after surgery. As a conclusion from this trial, the progastrin assay is a simple and inexpensive blood test that shows high diagnostic accuracy in patients with GI carcinomas, as well as promising therapeutic longitudinal changes across sequential managements. An ongoing assessment is being conducted to determine the value of progastrin as a multi-tumor screening assay and as a monitoring test [58].

The ONCORO study (NCT03787056) was a large case–control study designed to prospectively assess the hPG_80_ diagnostic and monitoring value in 420 patients with 16 types of cancer. In the study, the diagnostic value of hPG80 (primary endpoint) was assessed by comparing the hPG 80 baseline titers measured in cancer patients to those of 330 healthy volunteers who were recruited in parallel. The data relating to the different cohorts confirm that cancer patients have significantly higher blood concentrations of hPG_80_ than healthy subjects. Strong diagnostic accuracy was observed for lung cancers and hepatocellular carcinoma, with a specificity of >90%. It is interesting to note that the screening programs for certain types of cancers could potentially be improved with the use of a simple blood hPG80 ELISA assay. This could complement the limited specificity of imaging and provide a more effective way to detect these cancers early on [59].

## 6. Concluding Remarks

Defining accurate predictive biomarkers for targeted therapies and optimizing activation of the anti-tumor immune response are areas of an intensive ongoing investigation. As already mentioned, progastrin has a crucial role in tumorigenesis in a broad spectrum of human malignancies, including GI cancers. Firstly, hPG80 could function as a new circulating tumor biomarker on disease progression and evaluating the effectiveness of treatment. The idea of using a simple blood hPG80 assay to complement the limited specificity of cancer imaging is certainly intriguing. It could potentially enrich the screening programs for these cancers and help detect them at an earlier stage, which could ultimately lead to better treatment outcomes. It has been demonstrated that progastrin has the potential to be an innovative cancer target, which can be highly beneficial for cancer treatment. There are emerging therapeutic strategies that show promise in targeting progastrin; for example, blocking the interaction between progastrin and its receptor can inhibit cancer cell growth and improve outcomes in certain types of cancer, like the efficacy of RT in the treatment of CRC. Additionally, there is a need for more clinical studies to evaluate the safety and efficacy of progastrin-based therapies in humans and highlight even more clearly the role of progastrin as a biomarker. Therefore, further investigation to elucidate the complex interaction between progastrin and the immune system remains a significant challenge and an essential factor to consider in developing future strategies against cancer.

## Figures and Tables

**Figure 1 biomedicines-12-00885-f001:**
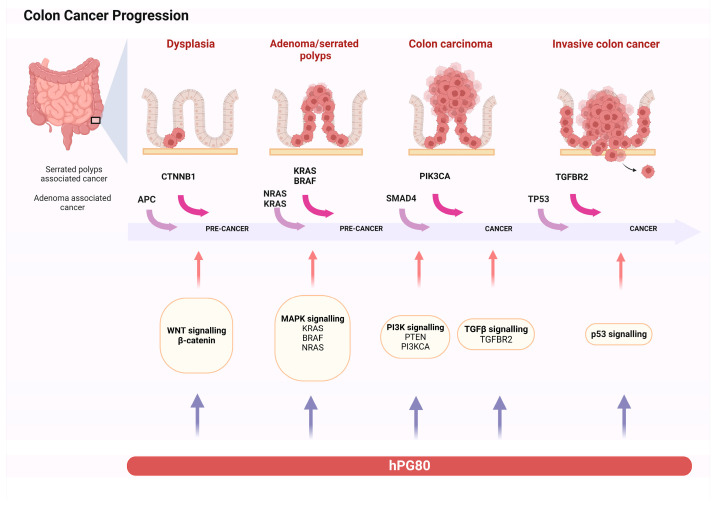
The figure shows a schematic representation of intestinal tumorigenesis, showing the accumulation of mutations, the activation of signaling pathways, and the role of progastrin (This figure was created based on the tools provided by Biorender.com).

**Figure 2 biomedicines-12-00885-f002:**
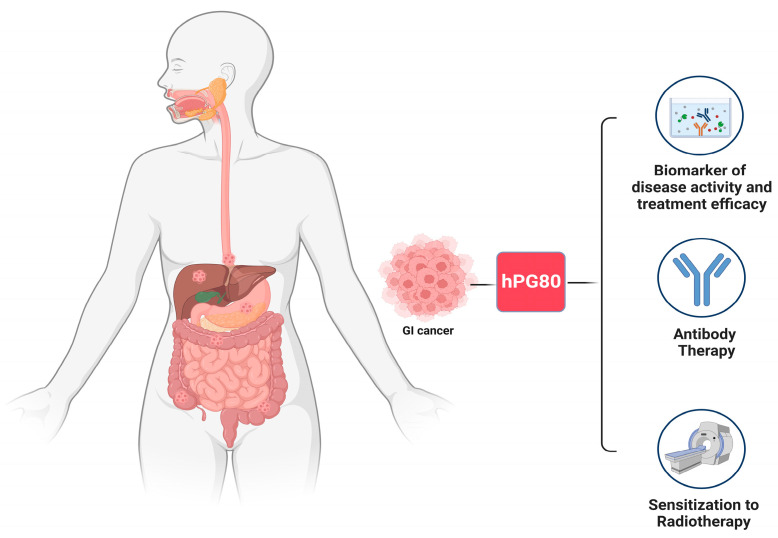
The figure shows the clinical implications of progastrin(this figure was created based on the tools provided by Biorender.com).

## Data Availability

Not applicable.

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
