# Peer review of "Progastrin: An Overview of Its Crucial Role in the Tumorigenesis of Gastrointestinal Cancers"

_biomedicines, 2024, doi:10.3390/biomedicines12040885_

Round 1

Reviewer 1 Report

Comments and Suggestions for Authors

The article provides a comprehensive overview of the clinical implications of Progastrin, covering its potential as a biomarker, antibody therapy, radiotherapy sensitizer, and ongoing clinical trials. It effectively highlights the importance of Progastrin in cancer management and presents valuable insights into its potential therapeutic applications.

However, there are a few areas where improvements could enhance the clarity and impact of the article:

While the article mentions ongoing clinical trials, it could benefit from a more focused discussion on the objectives, designs, and expected outcomes of these trials. Providing additional context and analysis would enhance the understanding of Progastrin's clinical potential.

Although the conclusion briefly touches on the importance of future research, it could be strengthened by further discussion of future directions in Progastrin research. This could include potential challenges, emerging therapeutic strategies, and areas requiring further investigation.

Comments on the Quality of English Language

The grammar and expression throughout the article are generally clear and fluent, enriched with a wide range of professional terminology and technical vocabulary.

Author Response

please find enclosed our REVISED manuscript entitled “Progastrin: an overview of its crucial role in tumorigenesis of gastrointestinal cancers” to be considered for publication. We would like to thank you and the reviewers for your thoughtful evaluation of our manuscript and for your most welcome comments/suggestions. Accordingly, we have now revised our manuscript thoroughly to reflect these comments.

Please find below a point-by-point response to ALL the issues raised by the Reviewers:

Reviewer #1:

The article provides a comprehensive overview of the clinical implications of Progastrin, covering its potential as a biomarker, antibody therapy, radiotherapy sensitizer, and ongoing clinical trials. It effectively highlights the importance of Progastrin in cancer management and presents valuable insights into its potential therapeutic applications.

However, there are a few areas where improvements could enhance the clarity and impact of the article:

While the article mentions ongoing clinical trials, it could benefit from a more focused discussion on the objectives, designs, and expected outcomes of these trials. Providing additional context and analysis would enhance the understanding of Progastrin's clinical potential.

Although the conclusion briefly touches on the importance of future research, it could be strengthened by further discussion of future directions in Progastrin research. This could include potential challenges, emerging therapeutic strategies, and areas requiring further investigation.

AUTHOR RESPONSE: Initially, we would like to thank the reviewer for his kind commentsand the fruitful remarks that will be beneficial for the current manuscript.We added additional information to the clinical trials section. We also expanded the conclusions on how progastrin can be used and the need for additional clinical trials.

Trusting that we have adequately addressed the Reviewers’ concerns, we would like to thank you for your help in improving significantly our work.

Kind regards,

Reviewer 2 Report

Comments and Suggestions for Authors

The study revealed that progastrin plays a significant role in the development of gastrointestinal cancers. It links progastrin to key signaling pathways involved in cancer progression and highlights its multifaceted role in cancer progression. The study concludes that progastrin has potential as a biomarker for disease monitoring and offers valuable implications for targeted therapies and treatment strategies. However, I have the following comments for revision:

1. The manuscript needs more detailed explanations of the research methodology and results. Also, it should provide more context about the significance of the research. Please clearly state the clinical impressions of your study in the last paragraph of the Introduction. What problems remain unanswered? What questions are you responding to?

2. Please provide more detailed information about the study's limitations and suggestions for further work.

3. The manuscript would be improved by including a section dedicated to the potential applications of the findings, which would help readers understand the significance of the study.

Author Response

Please find below a point-by-point response to ALL the issues raised by the Reviewers:

Reviewer #2:

The study revealed that progastrin plays a significant role in the development of gastrointestinal cancers. It links progastrin to key signaling pathways involved in cancer progression and highlights its multifaceted role in cancer progression. The study concludes that progastrin has potential as a biomarker for disease monitoring and offers valuable implications for targeted therapies and treatment strategies. However, I have the following comments for revision:

  1. The manuscript needs more detailed explanations of the research methodology and results. Also, it should provide more context about the significance of the research. Please clearly state the clinical impressions of your study in the last paragraph of the Introduction. What problems remain unanswered? What questions are you responding to?

AUTHOR RESPONSE:We thank the reviewer for the remark. This manuscript based on a bibliographic review. This is not a new study conducted by us to have research methodology and results.

  1. Please provide more detailed information about the study's limitations and suggestions for further work.

AUTHOR RESPONSE: We thank the reviewer for theconstructive comment.To the conclusions, we added what further can be done regarding progastrin and its role in cancer.

  1. The manuscript would be improved by including a section dedicated to the potential applications of the findings, which would help readers understand the significance of the study.

AUTHOR RESPONSE: We thank the reviewer for the apt comment. We added to the conclusion the potential applicationsand utility of progastrin.

Trusting that we have adequately addressed the Reviewers’ concerns, we would like to thank you for your help in improving significantly our work.

Kind regards,

Reviewer 3 Report

Comments and Suggestions for Authors

The paper is very interesting and it opens new windows about the research and monitor of gastro-intestinal neoplasms. Although, I suggest:

At page 8, line 303-304, the Authours should explain the role of Progastrin in precancerous lesions as Atrophic Gastritis and Intestinal Metaplasia both present in Autoimmune Gastritis and Severe Infection by Helicobacter Pylori

The Authors should speak about the role of Progastrin in Small Intestine and Colon Neoplasms and, if it is possible, to insert a little paragraph to page 9

Author Response

Please find below a point-by-point response to ALL the issues raised by the Reviewers:

Reviewer  #3:

The paper is very interesting and it opens new windows about the research and monitor of gastro-intestinal neoplasms. Although, I suggest:

At page 8, line 303-304, the Authours should explain the role of Progastrin in precancerous lesions as Atrophic Gastritis and Intestinal Metaplasia both present in Autoimmune Gastritis and Severe Infection by Helicobacter Pylori

The Authors should speak about the role of Progastrin in Small Intestine and Colon Neoplasms and, if it is possible, to insert a little paragraph to page 9

AUTHOR RESPONSE: Initially, we would like to thank the reviewer for his kind comments.

We thank the reviewer for the two remarks. In the literature, there is no direct association of progastrin with precancerous lesions; only gastrin has a direct association with precancerous lesions. Progastrin is directly involved only in cancer.

Regarding the second comment, we did not clarify that section 3 concerns CRC tumorigenesis; for this reason, we changed the title of paragraph 3.

Trusting that we have adequately addressed the Reviewers’ concerns, we would like to thank you for your help in improving significantly our work.

Kind regards,

Koustas Evangelos MD, PhD

Round 2

Reviewer 2 Report

Comments and Suggestions for Authors

Thank your for the revisions.